https://doi.org/10.1038/s41467-020-16435-x | OPEN

# Understanding the interface interaction between U$_3$Si$_2$ fuel and SiC cladding

Vancho Kocevski [1✉], Denise A. Lopes[1,2], Antoine J. Claisse[2] & Theodore M. Besmann[1]

Triuranium disilicide (U$_3$Si$_2$) fuel with silicon carbide (SiC) composite cladding is being considered as an advanced concept/accident tolerant fuel for light water reactors thus, understanding their chemical compatibility under operational and accident conditions is paramount. Here we provide a comprehensive view of the interaction between U$_3$Si$_2$ and SiC by utilizing density functional theory calculations supported by diffusion couple experiments. From the calculated reaction energies, we demonstrate that triuranium pentasilicide (U$_3$Si$_5$), uranium carbide (UC), U$_{20}$Si$_{16}$C$_3$, and uranium silicide (USi) phases can form at the interface. A detailed study of U$_3$Si$_2$ and SiC defect formation energies of the equilibrated materials yielding the interfacial phases U$_{20}$Si$_{16}$C$_3$, U$_3$Si$_5$ and UC reveal a thermodynamic driving force for generating defects in both fuel and cladding. The absence of either the U$_3$Si$_2$ or SiC phase, however, causes the defect formation energies in the other phase to be positive, removing the driving force for additional interfacial reactions. The diffusion couple experiments confirm the conclusion with demonstrated restricted formation of U$_3$Si$_5$, UC, and U$_{20}$Si$_{16}$C$_3$/USi phases at the interface. The resulting lack of continuous interaction between the U$_3$Si$_2$ and SiC, reflects the diminishing driving force for defect formation, demonstrating the substantial stability of this fuel-cladding system.

[1] Nuclear Engineering Program, University of South Carolina, Columbia, SC 29208, USA. [2] Westinghouse Electric Sweden, SE-72163 Västerås, Sweden. ✉email: vancho.vk@gmail.com

Maintaining the existing high standard of living in many countries, and allowing for improvement in the standard of living for others, will require a significant reliable supply of energy into the future. With the threat of global climate change, providing that energy from sustainable, low-carbon energy sources is at the forefront of energy research. As a mature technology, nuclear energy still remains at the leading edge as a near-term, scalable and low-carbon energy source[1,2]. However, the Fukushima-Daiichi accident raised issues of safety and security for nuclear facilities and materials, and addressing these issues has gained significant support, leading to international accident tolerant fuel (ATF) initiatives[3]. The main goal of an ATF is to provide an improved response to major failures in light water reactors (LWRs) leading to damaging temperatures, such as sustained loss of coolant or overpower-type accidents. The replacement of $UO_2$/Zr-alloy fuel was established as a principal goal in improving the accident behavior of the fuel-cladding system, ultimately reducing oxidation kinetics, heat of oxidation in steam, and release of hydrogen[4–6]. Thus, the key to ATF development is replacing the Zr-based cladding with a material less susceptible to steam oxidation and having a lower heat of oxidation. This in turn can require replacing the $UO_2$ fuel with higher uranium density phases, at least partially offsetting the increased parasitic neutron adsorption accompanying some of the new cladding concepts.

Due to its known resistance to oxidation and its low parasitic neutron adsorption cross-section, SiC/SiC composites have been considered as an ATF cladding option[6]. Correspondingly, the intermetallic fuel compound $U_3Si_2$ has received particular attention due to its combination of high thermal conductivity, reasonably high melting point, resistance to radiation-induced swelling and amorphization, and moderate oxidation resistance[7]. To assess $U_3Si_2$/SiC composite fuel cladding systems as an ATF option, understanding $U_3Si_2$/SiC interactions and thus their compatibility is paramount. This effort follows on work to refine and better understand phase equilibria and energetics in the U-Si system[8]. A significant contribution to such an assessment can be obtained from computational studies, notably using density functional theory (DFT). DFT has become a valuable tool, complementing experimental findings, providing a more fundamental understanding of observed phenomena, and in some cases, predicting new, useful materials or specific materials' behavior[9]. However, the potential of DFT for analyzing behavior at interfaces had not been fully recognized until the current efforts, focused on using the calculations to understand the atomistic scale driving forces behind interactions at the $U_3Si_2$/SiC interface. A broader aim of this study is thus to provide direction for using DFT in studying interface interactions among materials in general.

In this study, the phases that can be formed at the $U_3Si_2$/SiC interface were initially established from calculated reaction energies. $U_3Si_2$/SiC interactions were further investigated by computing defect formation energies, using the equilibrium phases determined to form at the $U_3Si_2$/SiC interface.

Considering that the interactions are driven by interdiffusion mechanisms enabled by defect formation, the calculated defect formation energies provide a way to understand and help predict the behavior at the atomic scale. The study of the compatibility of $U_3Si_2$ and SiC included exposing $U_3Si_2$ fuel to SiC in diffusion couples at well above likely operating temperatures, resembling severe accident conditions, ensuring interaction, and thus allowing for a thorough characterization of the interfacial zones. In addition, the phases that formed at the $U_3Si_2$/SiC interface were corroborated via characterization of pressed pellets of samples of equimolar $U_3S_2$ and SiC heated to high temperatures.

## Results and discussion

**Interface reaction energies**. The first step in understanding $U_3Si_2$/SiC interactions is identifying the phases that can form at the interface by finding the reactions with the most negative energies, $\Delta H_r$. Values of $\Delta H_r$ were calculated using:

$$\Delta H_r = \frac{1}{N_r}\left(\sum_i^p c_i \Delta H_f^i - \sum_j^r c_j \Delta H_f^j\right) \tag{1}$$

where $\Delta H_f^i$ and $\Delta H_f^j$ are the formation energies of the reactants, $i$, and products, $j$, respectively, and $c_i$ and $c_j$ are the reaction coefficients of the products and reactants, respectively; for more details on the formation energies the first-principles calculations section. The sums are over all products, $p$, and all reactants, $r$, and $N_r$ is the total number of atoms participating in the reaction.

We calculated $\Delta H_r$ values considering three binary and two ternary phases as products: USi, $U_3Si_5$, UC, $U_{20}Si_{16}C_3$, and $U_3Si_2C_2$. Note that because the $U_3Si_5$ phase has the most negative formation energy in the U–Si system[10,11], forming higher Si-content phases beyond $U_3Si_5$ would not be energetically favorable, and thus, were not considered. The $\Delta H_r$ of reactions yielding two product phases are shown in Table 1. Reactions generating more than two products result in zero degrees of freedom at fixed temperature and pressure, hence $\Delta H_r$ is not a point, but a continuous line representing a linear combinations of reactions with two products (Supplementary Fig. 1).

The reaction having the most negative $\Delta H_r$, reaction #2, forms $U_3Si_5$ and UC. At higher $U_3Si_2$ to SiC mole ratios, representing the fuel side of the fuel/cladding system, the $U_{20}Si_{16}C_3$ and USi phases form (#6), represented by the minimum $\Delta H_r$ at that composition on the $\Delta H_r$ convex hull (Supplementary Fig. 1). This implies that in addition to $U_3Si_5$ and UC, the pair of phases USi and $U_{20}Si_{16}C_3$ should also exist at the $U_3Si_2$/SiC interface proximate to the fuel surface. Reactions forming $U_3Si_2C_2$ have significantly more positive $\Delta H_r$ than competing reactions, eliminating its likely presence at the interface.

**Defect formation energies**. It is evident from the reaction energies that there is a driving force for interaction between $U_3Si_2$ and SiC. However, phase formation needs atomic transport and

**Table 1 Calculated $U_3Si_2$/SiC interface reaction energies.**

| # | Reaction | x | $\Delta H_r$ (eV/atom) |
|---|----------|---|------------------------|
| 1 | $5U_3Si_2 + 6SiC = 3U_3Si_2C_2 + 2U_3Si_5$ | 0.4545 | −0.1300 |
| **2** | $\mathbf{8U_3Si_2 + 9SiC = 5U_3Si_5 + 9UC}$ | **0.4706** | **−0.1785** |
| 3 | $3U_3Si_2 + 2SiC = U_3Si_2C_2 + 6USi$ | 0.6000 | −0.1202 |
| 4 | $2U_3Si_2 + SiC = 5USi + UC$ | 0.6667 | −0.1430 |
| 5 | $61U_3Si_2 + 27SiC = 9U_{20}Si_{16}C_3 + U_3Si_5$ | 0.6932 | −0.1397 |
| **6** | $\mathbf{7U_3Si_2 + 3SiC = U_{20}Si_{16}C_3 + USi}$ | **0.7000** | **−0.1386** |

$U_3Si_2$/SiC reaction energies, $\Delta H_r$, listed in increasing molar fraction, $x$, of $U_3Si_2$ in the reaction: $xU_3Si_2 + (1-x)SiC$. The reactions in bold represent the minimum $\Delta H_r$, i.e., $\Delta H_r$ on the reaction energy convex hull (Supplementary Fig. 1).

**Table 2 Calculated defect formation energies in $U_3Si_2$ and SiC.**

| Point defect in $U_3Si_2$ | Three-phase equilibria | | | | | |
|---|---|---|---|---|---|---|
| | $U_3Si_2$-SiC-$U_3Si_5$ | $U_3Si_2$-SiC-UC | $U_3Si_2$-SiC-USi | $U_3Si_2$-SiC-$U_{20}Si_{16}C_3$ | $U_3Si_2$-$U_3Si_5$-$U_{20}Si_{16}C_3$* | $U_3Si_2$-$U_3Si_5$-UC* |
| U1 vac. | 0.87 | 0.41 | 1.10 | **−10.27** | 0.87 | 0.87 |
| U2 vac. | 2.22 | 1.76 | 2.44 | **−8.93** | 2.22 | 2.22 |
| Si vac. | 1.13 | 1.82 | 0.79 | 17.84 | 1.13 | 1.13 |
| U-on-Si | 0.66 | 1.81 | 0.10 | 28.52 | 0.66 | 0.66 |
| Si-on-U1 | 0.40 | **−0.75** | 0.97 | **−27.45** | 0.40 | 0.40 |
| Si-on-U2 | 1.67 | 0.52 | 2.23 | **−26.19** | 1.67 | 1.67 |
| Si in site 2b | **−0.24** | **−0.93** | 0.10 | **−16.95** | **−0.24** | **−0.24** |
| Si in site 4h | 1.63 | 0.94 | 1.97 | **−15.09** | 1.63 | 1.63 |
| C-on-U1 | 1.29 | 1.52 | 1.18 | 6.86 | 3.15 | 2.44 |
| C-on-U2 | 2.27 | 2.50 | 2.15 | 7.84 | 4.12 | 3.42 |
| C-on-Si | 1.27 | 2.65 | 0.59 | 34.70 | 3.12 | 2.42 |
| C in site 2b | **−1.29** | **−0.60** | **−1.63** | 15.42 | 0.56 | **−0.14** |
| C in site 4g | 0.06 | 0.75 | **−0.28** | 16.78 | 1.92 | 1.21 |
| C in site 4h | **−0.24** | 0.45 | **−0.58** | 16.48 | 1.62 | 0.91 |
| C in site 8i | 6.42 | 7.11 | 6.08 | 23.14 | 8.28 | 7.57 |

| Point defect in SiC | $U_3Si_2$-SiC-$U_3Si_5$ | $U_3Si_2$-SiC-UC | $U_3Si_2$-SiC-USi | $U_3Si_2$-SiC-$U_{20}Si_{16}C_3$ | SiC-UC-$U_{20}Si_{16}C_3$* | SiC-$U_3Si_5$-UC* |
|---|---|---|---|---|---|---|
| Si vac. | 7.25 | 7.95 | 6.92 | 23.97 | 7.78 | 7.69 |
| C vac. | 4.82 | 4.13 | 5.16 | **−11.89** | 4.30 | 4.39 |
| Si vac. in Si-on-C** | **−0.05** | 0.64 | **−0.39** | 16.67 | 0.48 | 0.38 |
| U-on-Si | 4.34 | 5.49 | 3.78 | 32.20 | 5.49 | 5.49 |
| U-on-C | 11.18 | 10.95 | 11.29 | 5.61 | 11.27 | 11.47 |
| Si-on-C | 4.87 | 3.49 | 5.55 | **−28.56** | 3.81 | 4.01 |
| C-on-Si | 2.48 | 3.86 | 1.81 | 35.91 | 3.54 | 3.35 |
| U in site 4b | 12.88 | 13.34 | 12.66 | 24.03 | 13.50 | 13.60 |
| U in site 4d | 14.61 | 15.07 | 14.39 | 25.76 | 15.23 | 15.33 |

Defect formation energies ($\Delta E_f^D$) in eV/defect atom, in $U_3Si_2$ at chemical potentials governed by the indicated three-phase equilibria. Negative $\Delta E_f^D$ values are shown bold.
*Fuel and cladding not in contact.
**Si-on-C anti-site with vacancy on Si anti-site.

accumulation, requiring the generation of defects in $U_3Si_2$ and SiC. The energy for forming a defect, $\Delta E_f^D$, can be evaluated using:

$$\Delta E_f^D = E_{tot}^D - E_{tot}^0 - \sum_{i=1}^{N} \Delta N_i^D \left( \mu_i^0 + \Delta \mu_i \right) \qquad (2)$$

where $E_{tot}^0$ and $E_{tot}^D$, are the total energies of the supercell without and with a defect, respectively, and $\mu_i^0$ is the standard state chemical potential (DFT calculated energy per atom) of the element $i$. The standard state elements are α-U, diamond-Si and graphite. $\Delta N_i^D$ is the number of atoms of type $i$ added ($\Delta N_i^D > 0$) or removed ($\Delta N_i^D < 0$) from the perfect supercell to create the defect, and the sum is over all added and removed elements. $\Delta \mu_i$ is the change in the chemical potential of element $i$ resulting from an $N$-phase equilibrium (i.e., from the local environment), which can be calculated by solving the set of linear equations:

$$\Delta H_{f,k} = \sum_{i=1}^{N} c_{ik} \Delta \mu_i \qquad (3)$$

where the $\Delta H_{f,k}$ is the formation enthalpy of the phase $k$, and $c_{ik}$ is the mole fraction of element $i$ in phase $k$. From Eq. 3 it is evident that to determine the $\Delta \mu_i$, the number of phases $k$ and elements (components) $i$ should be equal, as there are zero degrees of freedom at constant temperature and pressure for three stable phases, completely defining the system.

The formation of any defect depends on the chemical potentials (Eq. 2), while the change in the chemical potentials, $\Delta \mu_i$, in turn, depends on the third phase formed from $U_3Si_2$ and SiC, establishing a 3-phase equilibrium (Eq. 3). In our case, we naturally consider $U_3Si_2$ and SiC to be in equilibrium, hence two phases are already determined, with the candidate third phase

being USi, $U_3Si_5$, UC, or $U_{20}Si_{16}C_3$. To model formation of an interface layer that separates the fuel and cladding phases, preventing them from equilibrating with each other, we allowed $U_3Si_2$ or SiC to be replaced with USi, $U_3Si_5$, UC or $U_{20}Si_{16}C_3$. In contrast to the routinely used approach of specifying that two bulk phases are stable[12,13], our approach allows studying the effect of the formation of a third phase at the interface. This method can also be used as a tool for identifying a phase or phases that could suppress the formation of defects in the phases in contact and hence, prevent interaction.

The calculated defect formation energies in $U_3Si_2$ and SiC from Eq. 2, are detailed in Table 2. The results for $U_3Si_2$-SiC-$U_{20}Si_{16}C_3$ demonstrate that there is a substantial driving force for forming Si defects in $U_3Si_2$; where Si can substitute for U on either of the two U sites and/or occupy one of two interstitial positions. In addition, formation of U vacancies on either U site is favored. Both Si incorporation and formation of U vacancies can lead to the formation of the Si-rich phases USi or $U_3Si_5$. In contrast, equilibrating $U_3Si_2$ and SiC with $U_3Si_5$, UC, or USi, (columns 1, 2, and 3 in Table 2) creates a driving force for inclusion of C on interstitial sites in $U_3Si_2$, potentially forming $U_{20}Si_{16}C_3$. There is also a driving force for Si incorporation on the 2b site when $U_3Si_5$ and UC are present, and Si on the U1 anti-site when UC is present. The negative defect formation energies when either of the three phases $U_3Si_5$, UC, and $U_{20}Si_{16}C_3$ are present implies that there is a substantial driving force for the formation of the Si-rich phase $U_3Si_5$.

The formation of Si vacancies was found to be not energetically favored for all studied 3-phase equilibria, thus $U_3Si_2$ will not decompose in contact with SiC to form a more U-rich phase. The formation energy for C interstitials is significantly negative, however, yet unlike Si, C cannot form anti-site defects in $U_3Si_2$.

Overall, there is a much higher preference for Si forming a U-Si-C ternary phase such as $U_{20}Si_{16}C_3$, as opposed to a C-rich phases such as $U_3Si_2C_2$. This conclusion agrees with the computed reaction energies that imply that $U_3Si_2C_2$ will not form at the interface (Table 1).

SiC in contrast, is understood to not tolerate Frenkel or Schottky defects, yet does display a wide variety of polymorphs based on stacking faults, which is again demonstrated in the positive incorporation and vacancy formation energies computed here (Supplementary Tables 2 and 3). Thus the $U_3Si_2$-SiC-$U_{20}Si_{16}C_3$ phase equilibrium-generated driving force for generating C vacancies in SiC, as well as Si-on-C anti-site defects, implies SiC decomposition. Decomposition product C will react with $U_3Si_2$, promoted by the equilibria, forming UC. Formation of the favored Si vacancy on a Si-on-C anti-site, as well as the driving force for forming Si vacancies, become the mechanism for forming a more Si-rich phase proximate to $U_3Si_2$. Ultimately, the net effect is the surface degradation of SiC at the contact interface.

Once a phase is formed on the fuel side such that $U_3Si_2$ and SiC can no longer directly interact, the governing three-phase equilibrium becomes either $U_3Si_2$-$U_3Si_5$-$U_{20}Si_{16}C_3$ or $U_3Si_2$-$U_3Si_5$-UC (columns 6 and 7 in Table 2). Under such conditions the defects have positive or only slightly negative formation energies interrupting any further interactions. The small negative value for Si on the 2b interstitial site allows diffusion of Si in $U_3Si_2$, promoting the formation of $U_3Si_5$. When $U_3Si_2$ is replaced in the equilibria by UC, generating the SiC-UC-$U_{20}Si_{16}C_3$ or SiC-UC-$U_3Si_5$ phase equilibria, defect formation energies are positive. Substituting USi for $U_3Si_5$ or $U_{20}Si_{16}C_3$, causes the considered defects to also have positive formation energies (results not shown), and were thus not considered. The implications of the defect formation energy calculations for the various equilibria are that after initial reactions between $U_3Si_2$ and SiC yielding interface phases, further interdiffusion would be very slow, essentially halting any further interactions. The only exception could be some transport of Si in $U_3Si_2$.

The defect formation energies suggest that when any of the three interface phases, $U_3Si_5$, UC, or $U_{20}Si_{16}C_3$ are present, there is a driving force for forming defects in both $U_3Si_2$ and SiC. On the other hand, the formation of any of these third phases requires a critical number of atoms to interact, the source of which are solely the formation of vacancies in $U_3Si_2$ and SiC. However, creating any vacancies, especially in SiC, requires substantial energy, such as high temperatures, thus serving as the bottleneck for initializing the $U_3Si_2$/SiC interaction. An analysis of the kinetics of the atoms at the $U_3Si_2$/SiC interface, such as their diffusivity over the interface, would provide a more thorough understanding of the influence of the temperature on the $U_3Si_2$/SiC interaction. In addition, the most complete picture of the driving force for $U_3Si_2$/SiC interactions can be obtained by looking at the defect formation energy at different types of $U_3Si_2$ and SiC surfaces forming a $U_3Si_2$/SiC interface. However, because the $U_3Si_2$ and SiC surfaces at the interface are not clearly defined, several possible interface combinations need to be considered, yet analysis of the entire number of possible defects would be prohibitive.

The results show that $U_{20}Si_{16}C_3$ phase encourages formation of U vacancies and Si incorporation in $U_3Si_2$ yielding $U_3Si_5$, on the fuel side. The ternary phase also promotes carbon vacancy formation and Si-on-C anti-site occupancy in otherwise very inert SiC, causing its decomposition and the formation of UC on the surface of the SiC. In turn, the $U_3Si_5$ and UC phases promote formation of Si and C vacancies in SiC and C interstitials in $U_3Si_2$, further supporting the formation of all three interface phases. When the $U_3Si_2$ and SiC are no longer in equilibrium due to an interface layer, there is no longer a driving force for most of the defects, and thus there is little observable interaction between the

fuel and cladding, with the exception of some formation of $U_3Si_5$ in the fuel.

**Diffusion couple experiments.** The computational results are reflected in the experimental observations for the $U_3Si_2$/SiC system diffusion couples annealed at 1200 °C/10 h and 1200 °C/100 h. The microstructures are shown in Fig. 1, where samples annealed at 1000 °C/100 h indicate no observable reaction, reflecting the results of differential scanning calorimetry (DSC) on the phase mixture (Supplementary Fig. 2). While the materials of the 1200 °C couples also did not adhere, easily separating during the disassembly of the jigs, an interdiffusion layer at the $U_3Si_2$ surface was observed with proximate Si-rich grains (Fig. 1a, b), reflecting the defect formation energies that allow incorporation of Si in $U_3Si_2$. The longer exposure time sample displayed $U_3Si_5$ grains beyond the inter-diffusion layer (Fig. 1b), indicating diffusion of Si in the $U_3Si_2$. This would naturally be the result of the calculated driving force for Si incorporation in $U_3Si_2$ occuring regardless of whether the $U_3Si_2$ and SiC are in contact.

No change in SiC composition was observed, with surface reactions causing some decomposition thus a concavity. SiC is relatively resistant to interdiffusing elements, and the observation reinforces the computed conclusion that the driving force for forming Si and C vacancies in SiC essentially decompose the surface. Microscopy images of the SiC surface and results of low angle x-ray diffraction (XRD) are seen in Fig. 2 where two different morphologies are observed: (i) large particles with a faceted surface (10–100 μm), typical of intermetallics such as $U_3Si_2$, and (ii) fine, dispersed particles, mainly found in cavities in the SiC surface. Both phases are uranium compounds as indicated by the bright contrast in the scanning electron microscopy (SEM) images. The larger particles were analyzed by energy dispersive spectroscopy (EDS) and had an average atomic composition of 52.7% (±2.8) U and 47.3% (±2.7) Si. This composition is close to USi, but it can also indicate the ternary $U_{20}Si_{16}C_3$ phase, as the low atomic number of C prevents it from being properly detected by EDS, and formation of either USi or $U_{20}Si_{16}C_3$ were shown to be possible. The fine, dispersed particles were too small to be analyzed by EDS, but low angle XRD yielded peaks that could be ascribed to β-SiC, $U_3Si_5$, and UC (Fig. 2c). The similar fcc structures of UC and SiC could encourage UC formation on the SiC surface, although their lattice parameters are markedly dissimilar. To further investigate the nature of these particles, high resolution SEM images were obtained (Fig. 2d) revealing the presence of spherical features adhered to the SiC surface. The spherical shape could reflect coherent growth on the SiC surface[14]. In addition, the phase is seen to be U-rich based on the bright contrast in the backscatter electron image.

To benchmark phase formation in the $U_3Si_2$-SiC system, annealed 1:1 molar mixtures of $U_3Si_2$:SiC powders were analyzed using DSC. The thermal trace of Supplementary Fig. 2a indicates an exothermic reaction occurring at ~1135 °C, supporting the lack of interactions observed in the diffusion couples annealed at 1000 °C. XRD analysis before the heating cycle showed major peaks for $U_3Si_2$ and SiC, with a minor amount of USi (FeB-type structure), the latter likely the result of the arc-melting synthesis of the original $U_3Si_2$ material. After heat treatment the XRD spectra indicate the presence of $U_3Si_5$, UC, and a minor amount of $U_{20}Si_{16}C_3$ (~6 mol%), in agreement with the reported U-Si-C phase diagram 11[15] and the computed reaction energies (Table 1). The results all suggest that the larger particles on the SiC surface are indeed the $U_{20}Si_{16}C_3$ phase (Fig. 2b).

In conclusion, a detailed investigation of the interaction between $U_3Si_2$ and SiC using DFT calculations of potential reactions and defect formation energies supported by experiment has delineated

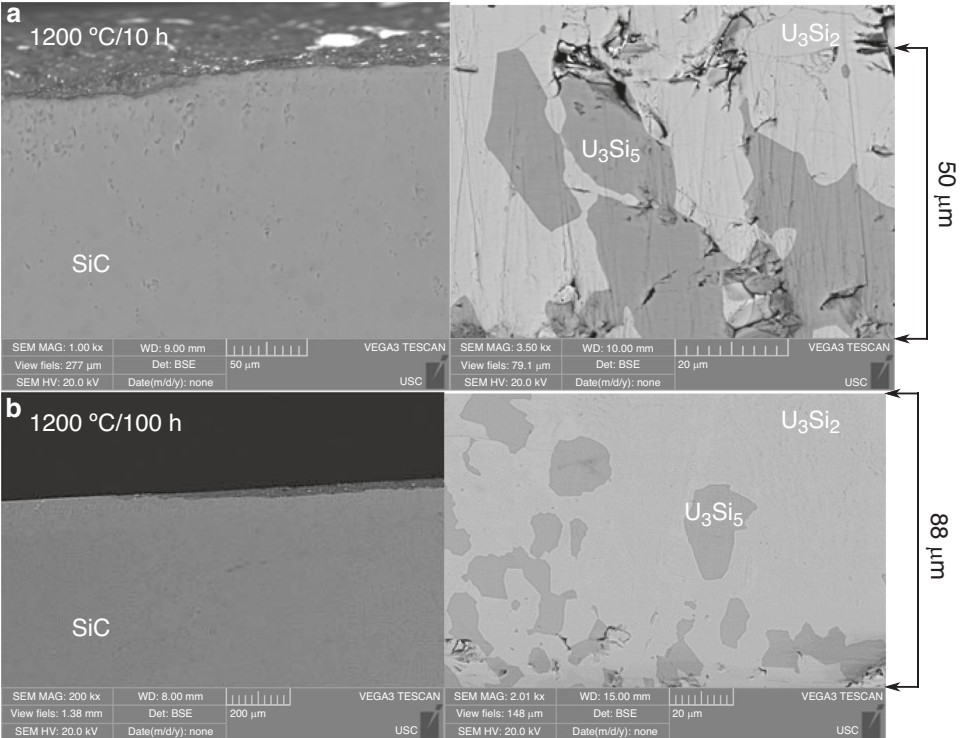

**Fig. 1 SEM analysis of U₃Si₂ and SiC.** Polished cross-section of the $U_3Si_2$ and SiC surfaces after exposure at 1200 °C for: (**a**) 10 h, and (**b**) 100 h. The depth of Si diffusion into the $U_3Si_2$ can be seen in the formation of $U_3Si_5$ well below the surface.

likely fuel-cladding interactions. Formation of $U_3Si_5$ and UC at the interface was shown to be favored, as was formation of $U_{20}Si_{16}C_3$ and USi on the $U_3Si_2$ fuel side. These phases allow formation of complementary defects in both $U_3Si_2$ and SiC, driving the interaction between the phases. Once an interaction phase is interposed between $U_3Si_2$ and SiC, the governing three-phase equilibrium becomes either $U_3Si_2$-$U_3Si_5$-$U_{20}Si_{16}C_3$ or $U_3Si_2$-$U_3Si_5$-UC. Under such conditions the defects have positive or only slightly negative formation energies interrupting any further interactions. The small negative energy value for Si on the 2b interstitial site allows diffusion of Si in $U_3Si_2$, promoting the formation of $U_3Si_5$, with defect formation energies becoming positive when $U_3Si_2$ is replaced by UC. The major implication of the defect formation energy calculations for the various equilibria are that after initial reactions between $U_3Si_2$ and SiC yielding interface phases, further interdiffusion becomes very slow, essentially halting any further interactions.

With regard to temperature dependence, the described interactions require formation of vacancies in $U_3Si_2$ and SiC, which need sufficient energy for their diffusion, explaining the absence of any observed interaction in the diffusion couples at 1000 °C. At the higher temperature of 1200 °C, $U_3Si_5$, UC, $U_{20}Si_{16}C_3$, and USi phases were observed at the interface, a result of limited interdiffusion, as implied by the DFT analysis. Thermal analysis of $U_3Si_2$:SiC equimolar mixtures indicated that $U_{20}Si_{16}C_3$ is stable in the phase region rather than USi. The cessation of fuel-cladding interactions due to the formation of a region devoid of either $U_3Si_2$ and SiC even at significantly elevated temperatures, argues for a very stable $U_3Si_2$ fuel/SiC cladding system and thus a potentially valuable ATF candidate.

## Methods

**First-principles calculations**. DFT calculations were performed with the Vienna Ab initio Simulation Package (VASP)[16,17]. The electron exchange correlation was modeled using the generalized gradient approximation (GGA) of Perdew, Burke and Ernzernhof (PBE)[18] and projector augmented wave (PAW) potentials[19,20]. To better describe the correlated nature of the U 5f electrons, we used the DFT + U

method with Dudarev's rotationally invariant approach[21] and an effective $U$, $U_{eff}$ = 1.5 eV ($U_{eff} = U – J$, $U$ = 1.5 eV and $J$ = 0.0 eV). The unit cells for $U_3Si_2$ and SiC were fully relaxed using a cut-off energy of 600 eV to expand the electronic wave functions. Convergence criteria of 0.01 eV/Å$^{-1}$ and $10^{-4}$ eV was adopted for the forces and total energy, respectively. We used $2 \times 2 \times 4$ and $3 \times 3 \times 3$ supercells of $U_3Si_2$ and SiC, respectively for calculating the defect formation energies. A γ-centered Monkhorst-pack k-point spacing of <0.02 Å$^{-1}$ was applied for sampling the Brillouin zone for each structure.

**Point defects**. The fuel/cladding interaction mechanism was further evaluated by calculating the formation energies of point defects in $U_3Si_2$ and SiC. The incorporation of C and Si in $U_3Si_2$ was modeled considering the two U sites, U1 (2a) and U2 (4h), the Si site, and 10 interstitials sites with Wyckoff positions: 2b, 2c, 2d, 4e, 4f, 4g, 4h, 8i, 8j, and 8k (Supplementary Fig. 3a and Table 1). Of the 10 interstitial sites, Si relaxed to only two sites 2b and 4h, while C relaxed to four sites 2b, 4g, 4h, and 8i. The incorporation of U and Si in SiC were modeled considering the Si and C sites, and two interstitial sites with Wyckoff positions 4b (0, 0, 0.5) and 4d (0.75, 0.75, 0.25) (Fig. 3b). Vacancies on U and Si sites in $U_3Si_2$, and Si and C vacancies in SiC were also considered.

Obtaining optimal total energies require using a $U_{eff}$ value that most appropriately represents the structure and the electron correlation in a system. For example, the properties of α-U are best represented by using only GGA[22], i.e., $U_{eff}$ = 0 eV, while it has been shown that the lowest $U_{eff}$ that best represent $U_3Si_2$ phonons is $U_{eff}$ = 1.5 eV[23]. However, because of the use of different $U_{eff}$ values, the resulting $\Delta H_f$ values cannot be directly compared. Therefore, to allow consistent comparison of $\Delta H_f$ for all phases, the methodology for correcting $\Delta H_f$ values proposed by Jain et al.[24] was applied, using experimentally reported formation energies[25]. Shown in Table 3 are the corrected formation energies. For defects where uranium is considered as potentially incorporated into SiC, an on-site correlation with $U_{eff}$ = 1.5 eV was applied.

## Experiments

**Materials**. The $U_3Si_2$ material for the current effort was prepared from depleted uranium (Los Alamos National Laboratory, 99.98%) and elemental silicon (Sigma Aldrich, 99.99%) by arc-melting[26]. Arc-melting was conducted inside a glovebox maintained at an oxygen and H$_2$O level <0.1 ppm. In addition, an in-line getter reduced the argon arc-melter purge gas to $10^{-10}$ ppm oxygen. The samples were arc-melted 5–6 times to ensure homogeneity. Chemical vapor deposition (CVD) SiC (β-phase) (Morgan Advanced Ceramics, Ultra-pure 99.9995%) was used in the diffusion couples as being representative of SiC-SiC composites as these are expected to be produced via chemical vapor infiltration (CVI) and thus will have a CVD SiC surface in contact with the fuel. For the experiments using mixed

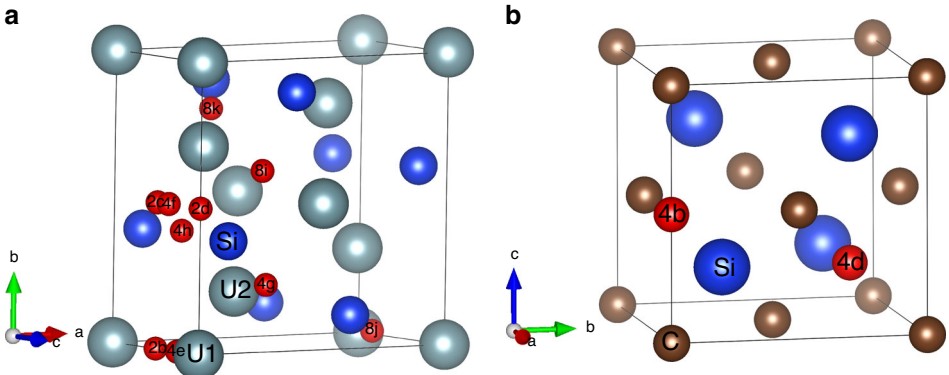

**Fig. 2 Analysis of SiC.** (**a**–**b**, **d**) SiC surface SEM images, and (**c**) low angle XRD, for a diffusion couple sample annealed at 1200 °C/100 h.

**Fig. 3 Position of interstitial sites in U₃Si₂ and SiC.** Interstitial sites (red spheres) in (**a**) $U_3Si_2$, and (**b**) SiC treated as point defects in DFT calculations. The U, Si, and C atoms are shown as gray, blue, and brown, respectively.

$U_3Si_2$-SiC powders, a commercial β-SiC powder (Alfa Aesar, 99.8% purity) was used with reported particle size of ~1 μm and surface area of a 11.5 m²/g. The $U_3Si_2$ powder was prepared by grinding in a tool steel mortar and pestle.

**Diffusion couples.** Sections ~2 mm in thickness of $U_3Si_2$, and ~1 mm in thickness of the CVD SiC were cut using a precision, diamond blade saw. The $U_3Si_2$ sample was ground and polished, using #600 and #1200 mesh silicon carbide paper. The CVD SiC was ground using diamond lapping plates of #600, #1200, #1800 mesh.

Subsequently, both samples were sequentially polished using 9 μm, 3 μm, 1 μm, and 0.25 μm diamond suspensions, with the final two polishing steps performed inside of a controlled atmosphere (<1 ppm $H_2O$ and <0.1 ppm $O_2$) glovebox to minimize the formation of an oxide layer. The $U_3Si_2$/SiC diffusion couple was assembled in the glovebox using a molybdenum jig (Fig. 4). Tantalum foil was interposed between the molybdenum plates and the samples to avoid potential reactions and the jig was wrapped in tantalum foil to getter residual oxygen. Immediately after assembling, the samples were transferred to a controlled atmosphere, resistance-heated tube furnace (CM 1730-12HT). The high temperature exposures were

**Table 3 Corrected formation energies of the studied phases.**

| Phase | Space group | Calculated $\Delta H_f$ | Experimental $\Delta H_f$ | Difference (calc. – exp.) | Calculation method |
|---|---|---|---|---|---|
| SiC | F-43m (216) | −0.2063 | −0.3465 | −0.1402 | GGA |
| $U_3Si_2$ | P4/mbm (127) | −0.3298 | −0.3538 | −0.0240 | GGA+U |
| USi | Pnma (62) | −0.4535 | −0.4335 | 0.0200 | GGA+U |
| $U_3Si_5$* | P6/mmm (191) | −0.4808 | −0.4480 | 0.0328 | GGA+U |
| UC | Fm-3m (225) | −0.4459 | −0.5055 | −0.0596 | GGA+U |
| $U_{20}Si_{16}C_3$ | Cmmm (65) | −0.4502 | −0.4450 | 0.0052 | GGA+U |
| $U_3Si_2C_2$ | I4/mmm (139) | −0.3733 | | 0.3733 | GGA+U |

Calculated formation enthalpies, $\Delta H_f$, in eV/atom, of U-Si-C binary and ternary compounds used for calculating $\Delta H_r$ and $\Delta \mu_i$, compared to tabulated values[25].
*The $U_3Si_5$ structure was taken from our previous cluster expansion study[11].

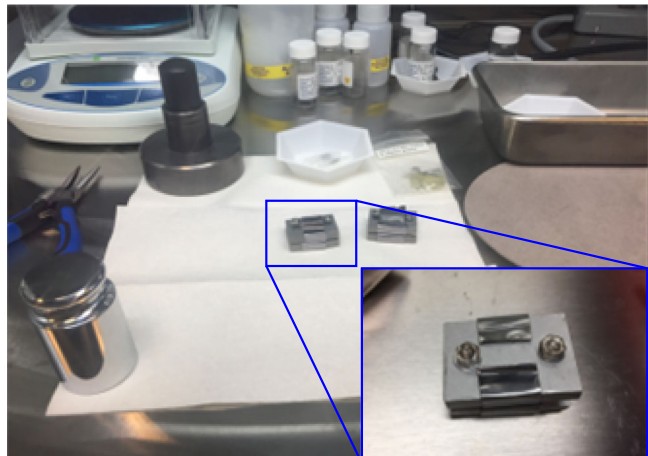

**Fig. 4 Diffusion couple setup.** Diffusion couple fixed in a molybdenum-jig inside a glovebox.

carried out for 10 or 100 h in flowing argon purified using a GEN'Air oxygen pump (SETNAG) to reduce the gas to $10^{-10}$ ppm $O_2$.

**DSC analysis**. The DSC samples were a 1:1 molar ratio mixture of $U_3Si_2$ and β-SiC powders as described above, with samples of ~1 g were manually blended and pressed into pellets. The studies were performed in a Netzsch STA-409 simultaneous thermal analyzer. The samples were heated to 1400 °C for 2 h at 5 °C/min under flowing purified argon, as previously described for the diffusion couples. The system was used in DSC mode and the thermal cycle was repeated twice to obtain a baseline for correction and detection of the onset of reactions. Characterization included SEM with phase composition determined by EDS using a Tescan Vega 3 SEM and a Zeiss Ultra plus field emission SEM (FESEM). Powder XRD analysis was carried out using a Rigaku Ultima IV instrument with Cu-Kα radiation and a scan over 20–100° at 0.02°/s. Rietveld analyses was performed using MAUD software[27,28].

## Data availability

The authors declare that the data supporting the findings of this study are available within the paper and its supplementary information files. Additional data that support the findings of this study are also available from the corresponding author upon reasonable request and archived on Open Science Forum https://osf.io/mdkzr/.

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

## Acknowledgements

This research was supported by the DOE Office of Nuclear Energy's Nuclear Energy University Programs. This work used the Extreme Science and Engineering Discovery Environment (XSEDE), which is supported by National Science Foundation grant number ACI-1548562, and the HPC Hyperion cluster supported by The Division of Information Technology at the University of South Carolina.

## Author contributions

V.K. and D.A.L. conceived the research idea, and performed the study. V.K. analyzed the DFT data, and interpreted the DFT and experimental results. D.A.L. generated the DFT data, and performed the experiments. The manuscript was written by Vancho Kocevski with contributions from D.A.L., A.J.C., and T.M.B. in the interpretation of the work. V.K., D.A.L., A.J.C., T.M.B. reviewed the manuscript.

## Competing interests

The authors declare no competing interests.
