## [Peer Review File · Nature Communications]

Reviewers' comments:

Reviewer #1 (Remarks to the Author):

This is an interesting and original paper. It provides new data on the fuel clad interaction in support with the development of accident tolerant fuels considering a concept with a SiC/SiC composite clad and a fuel compound U₃Si₂. Data provided are thermodynamic data obtained by both DFT calculations and experimental measurements.

The manuscript is clear and well presented. It provides new information and data presented are well supported and explained. Therefore I recommend publication.

I have however a few questions and comments to clear before publication.

1- Please check the English language throughout the manuscript.

2- The authors present interaction data between SiC and U₃Si₂ and calculations of reactions and defect formation energies supported by experiment. However, no information is given on the kinetics of these reactions. Experiments performed at 1000 °C for 100h showed no interaction. However, can the authors ascertain the fact that there is a critical temperature for initializing the SiC/U₃Si₂ interaction? Is it only a question of energy at this level or also a question of time?

3- At 1200 °C, the authors present results showing interaction between SiC and U₃Si₂ as also predicted by calculations. Only results obtained after 100 hours were presented. What would be obtained for longer duration? Is the kinetic modelling of the considered interaction mechanism is planned by the authors? Moreover, the authors say that due to the formation of phases at the SiC/U₃Si₂ interface, at a point, they are no longer in contact, therefore there are no reactions any longer except Si diffusion into U₃Si₂.

Is it really possible to conclude that in a real fuel/clad configuration there would no be any possibility to continue having some interactions? Is it possible to say that the formation is really homogeneous throughout the interface? Moreover concerning the Si diffusion, what would be its impact on longer duration? Could it be detrimental to the clad integrity?

4- Experiments were performed using CVD SiC, are more representative experiment using SiC/SiC composites with more representative geometries planned? Moreover are longer test duration considered?

Reviewer #2 (Remarks to the Author):

Reviewer report on the article "Understanding the interface interaction between U₃Si₂ fuel and SiC cladding" by Vancho Kocovski et al.

The manuscript "Understanding the interface interaction between U₃Si₂ fuel and SiC cladding" represents a combination of experiment and first principles calculation addressing chemical compatibility U₃Si₂ fuel and SiC cladding. Currently the U₃Si₂/SiC composite fuel/cladding system is considered a very promising option of international accident tolerance fuel initiative. To assess U₃Si₂/SiC as a possible candidate the chemical reaction between fuel and cladding are investigated using the density functional theory approach. Authors used this approach to establish phases formed at the interface from the calculated reaction energies. In addition the defect formation energies in U₃Si₂ and SiC in the presence of third phase have been calculated. The obtained information was used to understand U₃Si₅, UC and U₂₀Si₁₆C₃ phases formation at the atomic scale. Further, the results and their interpretations are supported by presented experimental observations. The analysis looks like a very interesting and promising approach to initial estimate of interactions, phases formation and general stability of complex interfaces in harsh environment.

In summary, this manuscript is of unquestionable interest for general auditorium of the Nature Communications readers. The story is well written and easy to read. I recommend publication in Nature Communication after some revision.

Major questions:

1. Equation 1, 2, 3 should be presented right at the place they are first mentioned, not at the end, in the Method section. To reviewer's opinion it will make the text much easier to read.
2. Authors discuss diffusion processes using information of the defects formation energies only. However, the defects propagation over the energy barriers separating two metastable or stable states is an important component of the diffusion process. It looks like some discussion here will be in place.
3. The defects formation energy discussed in the text are calculated in the bulk phases. While presence of different phases is incorporated through chemical potential. However, often the defects formation energy at actual (sharp, physical) interface is very different from the one in the bulk. I believe that, it will make the analysis stronger if the authors present a reason why these types of defects are not taken into consideration.

Reviewer #3 (Remarks to the Author):

Understanding the interface interaction between U_3Si_2 fuel and SiC cladding by Kocevski et al., examines the chemical interaction between U_3Si_2 fuel and SiC cladding by DFT computation and diffusion couple experiments. Calculation of reaction energies demonstrate that U_3Si_5 , UC, $U_{20}Si_{16}C_3$, and USi phases can form at the interface. Defect formation energies calculation reveal that a thermodynamic driving force for generating defects in both fuel and cladding exists when $U_{20}Si_{16}C_3$, U_3Si_5 and UC are in equilibrium together. Diffusion couple experiments were also carried out to document the formation of U_3Si_5 , UC, and $U_{20}Si_{16}C_3$ /USi phases at the interface.

While the subject matter of this paper could be important for the international accident tolerance fuel (ATF) program, the manuscript fails to provide any "understanding" beyond what one can gain from a simple assessment of U-Si-C ternary phase equilibria with some assumptions in atomic mobility – as far as potential phases that can form. In fact, the computational component of this work, while detailed in thermodynamics driving force and defect formation, omits the actual "diffusion kinetics" aspect. A better understanding would require, for example, knowledge of atomic mobility.

Documentation of experimental work is limited and confusing. Authors do claim that "The materials of the couples also did not adhere, separating during the disassembly of the jigs." Still it would be helpful to show, in one micrograph, SiC, U_3Si_2 and interaction layer in-between, so that the reader is spatially oriented.

Will the actual application of this fuel system be under high purity Argon? Any presence of oxygen (e.g., at the interface) would drastically change the thermodynamics and diffusion kinetics.

This manuscript is more suitable for Journal of Nuclear Materials, as it represents more of an incremental work, and does not provide comprehensive and/or transformative "understanding."

Reviewers' comments:

Reviewer #1 (Remarks to the Author):

This is an interesting and original paper. It provides new data on the fuel clad interaction in support with the development of accident tolerant fuels considering a concept with a SiC/SiC composite clad and a fuel compound U₃Si₂. Data provided are thermodynamic data obtained by both DFT calculations and experimental measurements.

The manuscript is clear and well presented. It provides new information and data presented are well supported and explained. Therefore I recommend publication.

I have however a few questions and comments to clear before publication.

We are very grateful to the referee for the valuable input and constructive criticism, and the prospect for future publication.

1- Please check the English language throughout the manuscript.

We thank the reviewer for the raised concern, and we edited the English language.

2- The authors present interaction data between SiC and U₃Si₂ and calculations of reactions and defect formation energies supported by experiment. However, no information is given on the kinetics of these reactions. Experiments performed at 1000 °C for 100h showed no interaction. However, can the authors ascertain the fact that there is a critical temperature for initializing the SiC/U₃Si₂ interaction? Is it only a question of energy at this level or also a question of time?

We are thankful to the reviewer for questioning the influence of kinetics on the U₃Si₂/SiC interaction. We like to bring reviewer's attention to our DSC experiment on equimolar mixture of powdered U₃Si₂ and SiC. In this experiment we show that there is indeed a critical temperature at which the U₃Si₂/SiC interaction starts, at 1135 C. Thus, supporting our observation that there is not interaction between the U₃Si₂ and SiC at 1000 C, only at 1200 C.

3- At 1200 °C, the authors present results showing interaction between SiC and U₃Si₂ as also predicted by calculations. Only results obtained after 100 hours were presented. What would be obtained for longer duration? Is the kinetic modelling of the considered interaction mechanism is planned by the authors? Moreover, the authors say that due to the formation of phases at the SiC/U₃Si₂ interface, at a point, they are no longer in contact, therefore there are no reactions any longer except Si diffusion into U₃Si₂.

Is it really possible to conclude that in a real fuel/clad configuration there would no be any possibility to continue having some interactions? Is it possible to say that the formation is really homogeneous throughout the interface? Moreover concerning the Si diffusion, what would be its impact on longer duration? Could it be detrimental to the clad integrity?

These are indeed very important questions, and we are grateful to the reviewer for bringing them up. Although, we do show that energy, by means of temperature, can initialize the reaction between SiC and

U₃Si₂, our analysis over time is for 10 and 100 hours. We like to point that although the chosen times might seem short, the chosen temperatures represent a severe accident conditions, at which Cr coatings fail in matter of minutes, while our system show great resistance for 100 hours, close to the 10 days selected by the ATF. Also, our results show that the only difference between the samples run for 10 and 100 hours is the amount of U₃Si₅ formed in the U₃Si₂ fuel, indicating that the diffusion process is mainly from the cladding to the fuel, causing only degradation of the cladding surface. Moreover, our focus is on understanding the phase equilibria and the atomistic mechanisms that generate the phase nucleation, with no intention of simulating operating fuel/cladding interaction mechanism. Therefore, we believe that having done experiments for up to 100 hours is sufficiently enough to demonstrate our main goal.

4- Experiments were performed using CVD SiC, are more representative experiment using SiC/SiC composites with more representative geometries planned? Moreover are longer test duration considered?

We thank the reviewer for noting the difference between the SiC in this study and the proposed SiC/SiC composite. The CVD SiC is more suitable for studying phase equilibria because it has a more regular surface than the real cladding SiC/SiC composite. As we previously mentioned, the main aim of this work is to study phase equilibria applying atomistic modeling coupled with experiments to understand the stages of the U₃Si₂/SiC interaction, and thus we are not focusing on the modeling fuel/cladding reaction in operation, and no further study using SiC/SiC is planned.

Reviewer #2 (Remarks to the Author):

Reviewer report on the article "Understanding the interface interaction between U₃Si₂ fuel and SiC cladding" by Vancho Kocovski et al.

The manuscript "Understanding the interface interaction between U₃Si₂ fuel and SiC cladding" represents a combination of experiment and first principles calculation addressing chemical compatibility U₃Si₂ fuel and SiC cladding. Currently the U₃Si₂/SiC composite fuel/cladding system is considered a very promising option of international accident tolerance fuel initiative. To assess U₃Si₂/SiC as a possible candidate the chemical reaction between fuel and cladding are investigated using the density functional theory approach. Authors used this approach to establish phases formed at the interface from the calculated reaction energies. In addition the defect formation energies in U₃Si₂ and SiC in the presence of third phase have been calculated. The obtained information was used to understand U₃Si₅, UC and U₂O₃ phases formation at the atomic scale. Further, the results and their interpretations are supported by presented experimental observations. The analysis looks like a very interesting and promising approach to initial estimate of interactions, phases formation and general stability of complex interfaces in harsh environment. In summary, this manuscript is of unquestionable interest for general auditorium of the Nature Communications readers. The story is well written and easy to read. I recommend publication in Nature Communication after some revision.

We greatly appreciate reviewer's recognition of the importance of our work, and the prospect for future publication.

Major questions:

1. Equation 1, 2, 3 should be presented right at the place they are first mentioned, not at the end, in the Method section. To reviewer's opinion it will make the text much easier to read.

We agree with the referee, putting Eq. 1, 2 and 3 where they are first mentioned will make it easier for the reader to follow the narrative and make the flow of the paper better. Thus, in the revised version of the manuscript we placed equations 1, 2 and 3 where they are first mentioned.

2. Authors discuss diffusion processes using information of the defects formation energies only. However, the defects propagation over the energy barriers separating two metastable or stable states is an important component of the diffusion process. It looks like some discussion here will be in place.

We thank the referee for guiding our attention to the importance of the diffusion of atoms at the interface. Indeed it is important to know the diffusion barrier because this will provide us with the information of how deep the atoms can move withing the materials starting from the interface. To obtain this information we can calculate the diffusion barrier over different types of interfaces. However, in the real U₃Si₂/SiC system, the U₃Si₂/SiC interface is not clearly defined, and hence there are many possible interfaces, ranging in sizes, that needs to be sampled to get a clearer picture of the diffusion. Furthermore, there might be different diffusion mechanism for the three types of atoms, which also need to be considered, and the models should be large enough to ensure bulk-like properties in the middle of the U₃Si₂ and SiC. Taking all this into account means that we should perform fairly large amount of calculations, for systems considered to be on the boundary of what DFT calculations can handle, making the whole study prohibitively expensive.

3. The defects formation energy discussed in the text are calculated in the bulk phases. While presence of different phases is incorporated through chemical potential. However, often the defects formation energy at actual (sharp, physical) interface is very different from the one in the bulk. I believe that, it will make the analysis stronger if the authors present a reason why these types of defects are not taken into consideration.

We are grateful to the referee for pointing out that defect formation energies will be different the interface of the two materials. Indeed, the defect formation energies will not only be different at the interface, they also depend on the distance from the interface. However, as we already mentioned, the U₃Si₂/SiC interface is not clearly defined, and thus we would need to consider several models of interfaces that could give us some information on the defect formation energies. Considering that defect formation energy should also be studied as a function from the distance from the interface and type of defect, such calculations would require substantial computational time, rendering the calculations unattainable with our current computational capabilities.

Reviewer #3 (Remarks to the Author):

Understanding the interface interaction between U₃Si₂ fuel and SiC cladding by Kocovski et al., examines the chemical interaction between U₃Si₂ fuel and SiC cladding by DFT computation and diffusion couple experiments. Calculation of reaction energies demonstrate that U₃Si₅, UC, U₂O₃Si₁₆C₃, and USi phases can form at the interface. Defect formation energies calculation reveal that a thermodynamic driving force for generating defects in both fuel and cladding exists when U₂O₃Si₁₆C₃, U₃Si₅ and UC are in equilibrium together. Diffusion couple experiments were also carried out to document the formation of U₃Si₅, UC, and U₂O₃Si₁₆C₃/USi phases at the interface.

While the subject matter of this paper could be important for the international accident tolerance fuel (ATF) program, the manuscript fails to provide any “understanding” beyond what one can gain from a simple assessment of U-Si-C ternary phase equilibria with some assumptions in atomic mobility – as far as potential phases that can form. In fact, the computational component of this work, while detailed in thermodynamics driving force and defect formation, omits the actual “diffusion kinetics” aspect. A better understanding would require, for example, knowledge of atomic mobility.

We are very thankful to the referee for the recognition of the importance of our work, and we appreciate the constructive comments. Indeed, an understanding of the atomic mobility over the U₃Si₂/SiC interface would provide deeper understanding of the interaction kinetics. However, performing such calculations would require substantial amount of computational time that makes the study prohibitively expensive. Furthermore, our study goes one step beyond what can be assessed by knowing the thermodynamics of the phases in the U-Si-C phase diagram, and provides an atomic level insight on the interaction between U₃Si₂ and SiC, which can be used as a model for future studies where preventing interface interaction is required.

Documentation of experimental work is limited and confusing. Authors do claim that “The materials of the couples also did not adhere, separating during the disassembly of the jigs.” Still it would be helpful to show, in one micrograph, SiC, U₃Si₂ and interaction layer in-between, so that the reader is spatially oriented.

We thank the reviewer for pointing out that our discussion of the experimental work might be confusing for the reader. Therefore, in the revised manuscript we made the part detailing the experimental work clearer. Also, we would like to point out that we are not able to show the actual layer between the U₃Si₂ and SiC, but in figure 1 we show both the U₃Si₂ and SiC samples, with their contact layers.

Will the actual application of this fuel system be under high purity Argon? Any presence of oxygen (e.g., at the interface) would drastically change the thermodynamics and diffusion kinetics.

Indeed, oxygen would dramatically change the whole picture, and it has been already shown that U₃Si₂ is susceptible to oxidation. However, this work does not intend to simulate fuel/cladding in operation condition, and thus, we focused only at the case where no oxygen is allowed to come in contact with the fuel/cladding system. If the cladding does not fail, the concentration of oxygen in contact with the

interface will be extremely small. Studying the influence of oxygen on the U₃Si₂/SiC system will be a continuation of our study, where we are modeling an operating or accident situation.

This manuscript is more suitable for Journal of Nuclear Materials, as it represents more of an incremental work, and does not provide comprehensive and/or transformative “understanding.”

We agree with the referee that this study indeed focuses on nuclear materials and thus, it is in principle suitable for Journal of Nuclear Materials. However, we would like to point out that this is a unique study, providing comprehensive understanding of the interaction between two materials on the basis of DFT starting from reaction energies, and going deeper by studying the defect formation energies, as well as supporting the finding with experimental observations. Such analysis can also be used for studying other system where interaction and phase equilibria play a significant role. Furthermore, the other two reviewers found that this manuscript provides new information and it is of importance for the general audience.

In summary, we have followed the suggestions of the reviewers to improve our manuscript. We sincerely believe that the revised version of our manuscript will be considered by the reviewers suitable for publication in Nature Communications.

REVIEWERS' COMMENTS

Reviewer #1 (Remarks to the Author):

This is the revised version of the manuscript following first reviews. The manuscript has been significantly improved even if some questions remain unanswered (impact of the duration of the tests for example on the interactions observed). However, as the authors explain the objectives of the paper are not to qualify a concept but to study phase equilibrium to understand U₃Si₂/SiC interaction. Surely many questions remain in the view of developing a concept. According to me, taking into account this remark, the manuscript is ready for publication.

Reviewer #2 (Remarks to the Author):

Reviewer report on the article "Understanding the interface interaction between U₃Si₂ fuel and SiC cladding" by Vancho Kocevski et al.

Authors addressed the concerns raised in previous review. I recommend the manuscript for publication in Nature Communication.

Reviewers' comments:

Reviewer #1 (Remarks to the Author):

This is the revised version of the manuscript following first reviews. The manuscript has been significantly improved even if some questions remain unanswered (impact of the duration of the tests for example on the interactions observed). However, as the authors explain the objectives of the paper are not to qualify a concept but to study phase equilibrium to understand U₃Si₂/SiC interaction. Surely many questions remain in the view of developing a concept. According to me, taking into account this remark, the manuscript is ready for publication.

We are very thankful to the reviewer for recognizing the value of our work, and for the raised constructive criticism of how this study can be expanded. We are also very grateful for the reviewer's recommendation for publication of our manuscript.

Reviewer #2 (Remarks to the Author):

Authors addressed the concerns raised in previous review. I recommend the manuscript for publication in Nature Communication.

We greatly appreciate reviewer's recognition of the importance of our work, and for recommending the publication of our manuscript.